# Effects of Three Interventions Combining Impact or Walking at Intense Pace Training, with or without Calcium and Vitamin Supplements, to Manage Postmenopausal Women with Osteopenia and Osteoporosis

**DOI:** 10.3390/ijerph191811215

**Published:** 2022-09-07

**Authors:** Carmen García-Gomariz, Celedonia Igual-Camacho, Enrique Sanchís-Sales, David Hernández-Guillén, José-M. Blasco

**Affiliations:** 1Departament de Infermeria, Universitat de València, 46010 Valencia, Spain; 2Group of Physiotherapy in the Ageing Process, Social and Healthcare Strategies, Departament de Fisioteràpia, Universitat de València, 46010 Valencia, Spain; 3Hospital Clínic i Universitari de València, 46010 Valencia, Spain

**Keywords:** menopause, osteopenia, osteoporosis, physical exercise, pharmacological treatment

## Abstract

The purpose was to assess the effects of three interventions on bone mineral density (BMD) to prevent the onset or progression of osteoporosis in postmenopausal women. Specifically, thirty-nine postmenopausal women, diagnosed with osteopenia or osteoporosis, implemented either high-impact training (G1), the same training + calcium and vitamin D intake (G2), or walked at an intense pace + calcium and vitamin D (G3). Baseline change (BC) in BMD was estimated using the femoral neck and lumbar spine T-scores. Participants were classified as having suffered fractures and/or falls before (24-month) and during the 2-year intervention. The participants—aged 61.8 years—were allocated into G1 (*n* = 9), G2 (*n* = 16), and G3 (*n* = 14). The groups evolved similarly over time; however, participants in G2 exhibited the largest T-score improvements with BC over 20%. G1 and G3 maintained BMD levels (BC = −7 to 13.3%; *p* > 0.05). Falls occurred similarly across the interventions, while the participants in G2 had the lowest percentage of fracture events (*p* = 0.037). Overall, the findings suggest that regular physical exercise may be effective in maintaining or improving BMD in postmenopausal women presenting with osteopenia or osteoporosis. Due to the limited sample size, the results are preliminary and warrant future randomized trials to validate the findings.

## 1. Introduction

Osteoporosis is a systemic skeletal disease characterized by low bone mass and micro-architectural deterioration of bone tissue, with a consequent increase in bone fragility and susceptibility to fracture [1]. Densitometric diagnostic criteria are usually based on a standardized T-score, which is presented as the number of standard deviations above or below the bone mineral density (BMD) of the healthy young population of the same sex. This parameter classifies the condition into three categories in terms of healthy young women: osteoporosis (<−2.5), osteopenia (−2.5 to −1), and normal (>−1) [2,3]. The term osteopenia refers to low bone density, and it differs from osteoporosis, in which the bones are weak and easily get fractured [4]. Osteopenia is also a marker of the risk of fractures. Individuals with osteopenia may or may not eventually present with osteoporosis, but they are certainly an at-risk population. Although this categorization was originally agreed by the World Health Organization (WHO), bone density should not be the only factor considered. A number of aspects of bone composition and structure, which are grouped under the term “bone quality”, also contribute to bone strength [5,6].

Most individuals with osteoporosis or osteopenia do not have other diseases that are responsible for their reduced bone health. Their condition results from an alteration in bone remodeling. Bone biomechanical properties are diminished (i.e., greater fragility and weakness due to lower bone mass), which can lead to an added complication: fractures. These occur with greater probability as a result of a fall event, although they can also occur during other activities or even spontaneously [7,8]. As for the loss of bone mass, it is associated with two main factors: age and menopause. Concerning the latter, accelerated bone loss is closely related to the cessation of ovarian function and estrogen deficiency. For this reason, postmenopausal women are more prone to developing osteoporosis and therefore, suffering from fractures [8].

The prevention and treatment of osteoporosis have traditionally focused on pharmacological and non-pharmacological measures [9]. The progressions made in pharmacological treatment have been essential for reducing the loss of BMD or promoting bone formation through different mechanisms. Bisphosphonates, raloxifene, estrogens, and antiresorptive drugs are commonly combined with the adequate intake of calcium and vitamin D [10,11]. In addition, physical exercise is a feasible treatment [12,13,14]. Diverse approaches have been evaluated, ranging from impact and high-impact exercises to resistance training, vibrating platforms, endurance training, or walking [13]. Overall, the evidence supports that exercise contributes to the strengthening of bones [15] since it can maintain bone density [16] and has a positive impact on bone structure if the intensity of the exercise is higher than the minimal effective strain threshold of the bone [17]. Some reviews seem to suggest that combined exercise (aerobic, strength, flexibility, and impact or resistance exercise) may be more effective in preventing osteoporosis than standard care (usual activity or placebo) [12,18,19].

Previous studies in postmenopausal women have paid attention to fall and fracture incidence [20,21] as well as fall prevention [21,22,23]. As for the studies assessing the effects of physical exercise, these mostly focused on BMD or other outcomes such as C-reactive protein [21,24]. Some of them also monitored falls, while the impact of exercise on fractures has been monitored to a lesser extent [12,25]. One of our previous of randomized trials examined the effects of two types of intervention (high-impact and walking at intense pace) on BMD in postmenopausal women [13]. In this study, we present a secondary analysis to assess the effects of such interventions, combined with or without bone-enhancing vitamin supplements; in this case, the eligible sample was postmenopausal women with osteopenia and/or osteoporosis alone. Accordingly, the goal was to assess the effects of three interventions (high-impact training, high impact + calcium and vitamin D, and walking at an intense pace with calcium and vitamin D) on BMD, and to monitor fall and fracture events in postmenopausal women with osteopenia or osteoporosis.

## 2. Materials and Methods

### 2.1. Methods

The methods and compliance with the ethical standards of the Declaration of Helsinki were reviewed and approved by the Comité Ético y Científico of the Centro de Salud Rafael Romeu, Enguera, Spain (no. 12-03/2012). Participants were informed verbally and in writing, and they signed a consent form prior to participation. The research team consisted of a physiotherapist, who was not blinded to the interventions and was responsible for the design and recruitment; a medical doctor, blinded to group allocation, who recruited the sample and prescribed the densitometry; another member (a nurse), who was in charge of supervising the interventions and did not take part in the allocation process; and an external advisor (a physiotherapist), who was blinded to the group allocation and was in charge of data collection and outcome assessment. All health and safety procedures were performed during the study.

The sample included postmenopausal women over 55 years of age who presented with osteopenia or osteoporosis, verified with a diagnosis at the level of the femoral neck or lumbar spine (T-score < −1.0). Women who were prescribed medications for osteoporosis other than calcium or vitamin D, such as bisphosphonates, were not eligible for inclusion. Potential participants were recruited on the health center where the study was conducted. Information brochures explaining the study were provided to the health center. There were also posters on informative talks and advertisements. This was a secondary analysis, so an a priori formal sample size power calculation was not performed.

Of the 66 women who originally signed the informed consent agreeing to participate, 13 were excluded because they did not meet the inclusion criteria; overall, 53 were included, and those with BMD values within normality were excluded, while one participant dropped out when diagnosed with cancer. Data from 39 women were finally analyzed, of which 9 participants were included in G1, 16 in G2, and 14 in G3. (see Figure 1). The diagnosis of osteopenia or osteoporosis for each participant has been provided in the Appendix A. Participants were placed into groups according to the different interventions: Group 1 (G1; exercise only), Group 2 (G2; exercise plus calcium and vitamin D supplements), and Group 3 (G3; walking at an intense pace plus calcium and vitamin D supplements). Regarding group allocation, it was not possible to randomize the entire sample in origin because the calcium and vitamin D supplements were medically prescribed for the participants before they took part in the research, and this factor could not be controlled. Based on this limitation, the original allocation was performed as follows: eligible postmenopausal women who did not take vitamin supplements were allocated to G1, whereas those who took calcium and vitamin D supplements participated in a randomized clinical trial (NCT03091088), and they were randomly allocated (random number generator, i.e., Matlab software) to G2 and G3. The results of this randomized clinical trial have been previously published [13]. The main difference between this and such a manuscript is that only patients diagnosed with osteopenia and osteoporosis have been analyzed, while falls and fractures are also analyzed as outcomes of interest.

### 2.2. Interventions

The types of exercise to increase BMD have been classified in four main categories [26]: (1) strength and/or resistance training, in which the joints are moved against some kind of resistance, such as free weights, machines, or bodyweight [27,28]; this could include an impact component; (2) weight-bearing aerobic exercises, such as impact training or any other exercise in which the arms, feet, and legs bear the weight. Other common approaches include (3) multicomponent exercises (e.g., aerobics, dancing, and progressive resistance), whose effects have been difficult to quantify, and (4) whole-body vibration, which uses dedicated devices; these are effective for enhancing muscular strength; however, the effects on BMD remain controversial [26].

Considering the aforementioned characteristics, in this study, the interventions were based on the first two categories. The interventions lasted for two years, with 46 weeks planned per year. The intervention for G1 included resistance and high-impact training three times per week. The sessions had three parts and began with a warm-up phase, which was initiated with an informative talk about the session, and it included joint mobilization; muscle activation exercises; coordination and balance exercises, for instance walking sideways; walking backwards; walking on tiptoes; and so on (10 to 15 min, one exercise per min). The work phase or core phase had two parts, including 15 min of exercise, which followed the recommendations for exercise of the Asociación Española con la Osteoporosis y la Artrosis (Spanish Osteoporosis and Arthritis Association) for preventing osteoporosis in postmenopausal women, and 15 to 20 min of training was performed using such resources as weighted balls and elastic bands, which included maximum strength exercises by lifting free weights (resistance training) and moderate to high-impact exercises, by putting a high level of impact on the joints (e.g., exercises involving jumps, elastic bands, Pilates balls, dumbbells, etc.). In the third phase, relaxation techniques were included, aided by breathing exercises, massage, self-massage, and stretching. 

The participants in G2 implemented the same training as G1, with the intervention difference being that participants in G2 were medically prescribed calcium and vitamin D supplements (the dosage is described later in this section). 

The participants in G3 trained by walking at an intense pace and took calcium and vitamin D supplements. The training involved low to moderate impact by covering a distance of 6 km in 1 h [29]. This training had to be implemented three to five times per week, and it was proposed that different routes were used in natural places that surround the town. Participants were allowed to complete some alternative route though. Keymaze 700 Kalenji pedometers (Decathlon, Villeneuve d’Ascq, France) were provided to each participant to register the times and distances.

As for the bone-enhancing supplements used by G2 and G3, most of the participants (G2, *n* = 15; G3, *n* = 13) were prescribed a dosage of 1500 mg of calcium carbonate (600 mg calcium) and 400 IU of cholecalciferol (vitamin D3). Only two participants (one in each group) were prescribed a dosage of 2500 mg of calcium carbonate (1000 mg calcium) and 880 IU of cholecalciferol (vitamin D3).

The sessions for G1 and G2 were supervised at the health center, and the compliance was electronically registered using an access card. To verify G3’s compliance, participants were provided with a calendar to register the sessions, times, and distances. Additionally, a physiotherapist made weekly phone calls to ensure compliance. 

A maximum loss of 20% of session attendance was assumed to consider the intervention as satisfactory.

### 2.3. Measures

BMD was measured using a T-score index, at the beginning of the study after 2 years of intervention. Measurements were taken using dual energy X-ray absorptiometry (DXA), performed using a GE Lunar DPX Pro X-ray densitometer (GE Healthcare, Chicago, IL, USA). The locations were the lumbar spine and femoral neck since these levels are accepted by the World Health Organization and have been recognized as being the most reliable for the diagnosis of osteoporosis (World Health Organization 1994 Geneva: WHO). 

Fall was defined as the event in which a subject unintentionally came to rest on the ground or at some lower level and not a result of a major intrinsic event (such as a stroke) or overwhelming hazard [30,31]. Participants were provided with a calendar to record any possible fall event during the intervention (24 months). Participants were periodically reminded (phone calls every four months) to fill the calendar and were asked about the number of fall events with the question: From the last time we spoke, have you had any falls, including a slip or trip in which you inadvertently lost your balance and landed on the ground or lower level? A register of previous falls and fractures (24 months) was created by direct questioning about their occurrence before the interventions were initiated. Additionally, previous fracture events were extracted from the patients’ clinical histories. Finally, a one-minute osteoporosis risk test was implemented at baseline. This measure was created by the international osteoporosis foundation [32], which consists of a 19-item questionnaire that calculates osteoporosis risk factors based on questions about family history, clinical information, and lifestyle information. 

### 2.4. Data Analysis

Descriptive statistics were used for the analysis of demographic characteristics and risk factors for osteoporosis. The normality of the distribution and variance homogeneity of the quantitative variables was verified using the Shapiro–Wilk and Levene tests. Between-group differences at baseline regarding the T-score index of the femoral neck and lumbar spine, age, and body mass index were evaluated using *t*-tests. An analysis of variance based on a mixed-linear model was used; the main effects for time, group, and time per group were evaluated in terms of BMD, as measured using the T-score. The Tukey test was used in post-hoc analysis; partial eta squared (η^2^) results up to 0.01, 0.06, and 0.14 indicated a small, medium or a large effect, respectively. For data-analysis purposes, participants were classified into those who had fallen at least once or those who had not fallen (fallers vs. non-fallers). The same classification was used for fractures. Data was analyzed using chi-square tests to determine group differences in fall and fracture events (yes/no) at baseline and at the end of the 2-year intervention. Cramer’s *V* was used as a measure of the strength of the association, and a value over 0.5 was interpreted as a strong association [33]. All confidence intervals were set at a 95%. Raw data have been provided as Appendix A to this manuscript, for transparency.

## 3. Results

Thirty-nine participants with an average age of 61.8 were analyzed. There were 9, 16, and 14 participants in G1, G2, and G3, respectively. The flow of the participants is illustrated in Figure 1, and their characteristics are presented in Table 1. There were no baseline differences in the number of fallers or participants with previous fractures. There were no baseline differences in their T-scores, demographic characteristics, or any other osteoporosis risk factors than being smokers (3 participants in G3 were smokers at baseline, but all participants gave up during the intervention). Compliance with interventions exceeded 85% in all the groups. To our knowledge, no adverse events were associated with the intervention.

### Effects of the Interventions

Change in the T-score from baseline suggested that there was a significant time effect for the lumbar spine (*p =* 0.041), but not for the femoral neck (*p =* 0.259). Specifically, the post-hoc analysis suggested that participants in G2 (high impact + supplements) experienced the largest average baseline change (BC) (>20%), although this result was only statistically significant for the lumbar spine (*p =* 0.017). Participants in G3 (intense pace + supplements) maintained their BMD levels (BC = 6% to 13.3%; *p* > 0.05), as did participants in G1 (impact training alone) (BC ≈ −7%; *p =* 0.05). There was no significant time per group interaction, but the measure of the group effect size was moderate in the femur (η^2^ = 0.076) and lumbar spine (η^2^ = 0.12). The detailed information is provided in Table 2. 

The results suggested that there was a similar number of falls regardless of the type of intervention since no significant between-group differences were found in the number of fallers versus non-fallers (*p =* 0.213). However, the analysis revealed that G1 was the only intervention without fallers. As for fractures, G3 had the greatest number of participants who suffered fracture events (*p =* 0.037), as presented in Table 3. The results indicated that G2 had the lowest proportion of participants with fractures (1 out of 16, 6.3%).

## 4. Discussion

This study had two major findings. First, there was no difference in progression over the 2-year intervention period in terms of BMD when measured at the femoral neck or lumbar spine; therefore, we deduce that the studied interventions are similarly effective. Second, participants maintained or even improved their BMD levels after 2 years, regardless of calcium and vitamin D intake or type of exercise. Both findings support the view that physical exercise is a feasible approach to maintaining or even improving bone density in postmenopausal women who already present with osteopenia or osteoporosis. While these results indicate that the performance of high-impact and resistance training along with taking bone-enhancing supplements for the prevention of osteoporosis may be more beneficial, as one of our trials already indicated [13], these are the results of a secondary analysis, and the limited sample size and the impossibility of randomization of G1 means that future research is needed to verify this statement. 

It is known that BMD decreases gradually after the fourth decade of life, and efforts should be focused on weakening this process; maintaining the levels; and avoiding undesired consequences, such as fractures, especially in at-risk populations. Bone-enhancing medications are considered essential [10,11]. Regarding the effects of physical exercise combined with bone-enhancing supplements, previous research has assessed various approaches. Bolton et al. (2012) [21] suggested that exercise appears to have modest benefits for postmenopausal women with osteopenia who are not using drugs, while the extensive review conducted by Howe et al. (2011) [12] concluded that there is a small but possibly important effect of exercise on bone density when compared with controls. While moderate effects have been attributed to the practice of physical exercise alone, some other studies have suggested that the optimal approach is possibly a combination of reducing the risk factors, performing physical exercise, and providing pharmacological treatment [1,34], which is consistent with our results. However, the literature is still inconsistent, and some authors support pharmacological treatment as the key solution [11]. 

Regarding the most suitable type of exercise, most studies have concluded that combined exercise interventions that include high-impact, strength, flexibility, and resistance work are more effective than standard treatments for maintaining usual activity or receiving a placebo [35,36]. The same has been observed in the present study because regardless of exercise, all groups have shown positive effects on BMD after the intervention. Previous studies that have evaluated the effects of walking at an intense pace are scarce; nevertheless, the literature seems to suggest that this approach would also be suitable for managing osteoporosis, which is reinforced by our results. However, our research also found that high-impact and resistance training combined with calcium and vitamin D supplements had the greatest impact on conventional BMD evaluations. A limitation to this statement is that it is based on a qualitative assessment of the results and is up for speculation since this group did not present statistically significant differences compared to the other groups. Therefore, we cannot attribute greater benefits to this proposal than walking at an intense pace. 

Although postmenopausal women who present with osteopenia or osteoporosis are prone to accelerated BMD reduction, in this study, bone density was maintained in the femoral neck and improved in the lumbar spine over time. The reasons why the improvements were greater at one of the two levels are unclear, but this finding would suggest the important role that exercise plays in controlling bone mass loss and indirectly reducing the risk of fracture. One of the groups had no supplements, and positive results.

Due to several pathogenic mechanisms, there is an increase in bone fragility and the likelihood of fractures with ageing. Our study found that high-impact training, together with calcium and vitamin D supplements, may be effective in increasing BMD, and, although indirectly, this could contribute to a reduction in the risk of fractures. This is consistent with a previous study in which no adverse effects or fractures were observed in women who underwent impact training [37]; however, the study did not assess pharmacological treatment. Senderovich, Tang, and Belmont [38] similarly reported that long-term regular exercise, along with an increased vitamin D and dietary calcium intake, is most effective for preventing osteoporosis and reducing fractures. In line with this, we also found that almost 90% of our sample did not fall frequently. This could suggest that physical exercise may have an impact on fall prevention, which is widely supported by a large number of studies whose main objective was to assess falls in older adults [39].

A practical implication of our findings is to support the performance of exercise to increase BMD. The DXA of the calcaneus or phalanges, or other techniques, such as computed tomography or risk assessment tools, have provided inconsistent results with non-unified criteria. However, a review that examined 16 trials suggested that the femoral neck and lumbar spine T-scores are the most frequently reported and reliable tests for BMD [40], which indicates the practical value of the findings. Therefore, identifying the optimal way to increase BMD is a good approach for preventing fatal consequences. However, many authors have already advised that the global burden of the consequence of osteoporosis means that assessing the risk of fractures should also be a high priority amongst the health measures considered by policymakers [41], which supports the importance of implementing fracture risk assessment strategies worldwide. 

This study had several limitations. The sample size was not estimated a priori and the possibility of statistical errors could be considered; also, the sample was limited to those who were willing to participate, and only postmenopausal women with osteopenia and osteoporosis were included. This was the main reason to present preliminary results, which should be interpreted with utmost caution. Despite this, we found an effect on BMD that was moderate in the spine, so this trial may have provided useful information for future research endeavors. Regarding group allocation, it was not possible to randomize the entire sample because the calcium and vitamin D supplements were medically prescribed for some of the participants before they took part in the research, and this factor could not be controlled; therefore, eligible postmenopausal women who did not take vitamin supplements were allocated to G1, and those who took calcium and vitamin D supplements were randomly allocated to G2 and G3. Furthermore, it would have been interesting to conduct the densitometry more frequently for each participant at intermediate moments during the two years of the interventions; this would have allowed us to register effects at intermediate time points. We have not discussed the effects of pharmacotherapy alone or the implications of the risk factors in-depth because none of the groups received bone-enhancing supplements only and the participants’ habits were generally healthy. On the other hand, there is a growing interest in the effects of nutritional recommendations, which can be taken into consideration to guide the design of future studies. Importantly, adequately powered trials are needed to support the findings.

## 5. Conclusions

The findings support that regular physical exercise is an effective approach for maintaining BMD in postmenopausal women with osteopenia or osteoporosis since the participants overall maintained or improved their BMD levels regardless of group allocation. In addition, the results seem to suggest that high-impact and resistance training along with bone-enhancing supplements, such as calcium and vitamin D, might be the most effective intervention—among those we have studied at least—for preventing the onset and progression of osteoporosis. Due to limited sample size, adequately powered randomized trials are needed to support and confirm the findings.

## Figures and Tables

**Figure 1 ijerph-19-11215-f001:**
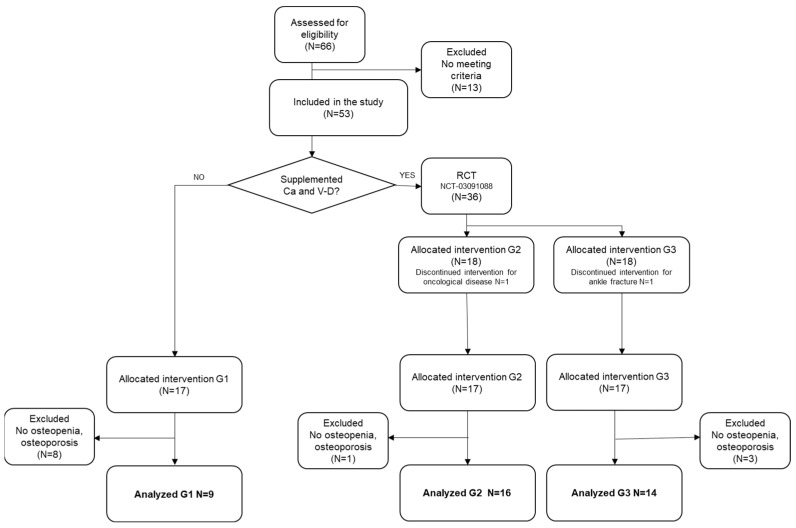
Flow chart of participants.

**Table 1 ijerph-19-11215-t001:** Baseline characteristics of the study sample.

	Baseline						*p*-Value
	G1 (*n* = 9)		G2 (*n* = 16)		G3 (*n* = 14)		
Sample descriptors							
Menopausal age (years)	51.0 (2.7)		49.4 (5.2)		51.1 (4.9)		0.56
Age (years)	60.3 (6.9)		64.9 (7.1)		59.4 (6.3)		0.07
Body mass index BI (kg/m^2^)	25.7 (1.9)		25.8 (3.1)		27.4 (4.6)		0.39
Weight BI (kg)	65.5 (8.5)		62.3 (7.4)		63.5 (7.4)		0.80
Weight AI (kg)	63.4 (8.3)		63.6 (7.9)		62.7 (7.0)		0.29
T-Score							
Femoral neck	−0.756 (0.746)		−0.786 (0.656)		−1.079 (0.830)		0.47
Lumbar spine	−1.479 (0.254)		−1.913 (1.061)		−2.307 (0.793)		0.08
Risk factors osteoporosis	Yes	No	Yes	No	Yes	No	
Family history osteoporosis	1	8	2	14	3	11	0.73
Previous fractures	1	8	1	15	1	13	0.90
Frequent falls	2	7	4	12	3	11	0.97
Body mass index < 18.5 kg/m^2^	0	9	1	15	1	14	0.72
Corticosteroids	0	9	1	15	1	13	0.72
Rheumatoid arthritis	0	9	1	15	1	13	0.72
Hyperthyroidism	1	8	1	15	3	11	0.20
Early menopause (<45 years)	0	0	2	14	2	12	0.50
Amenorrhea (>12 months)	0	9	1	15	2	12	0.43
Ovarian removal (no treatment)	1	8	1	15	3	11	0.45
Alcohol (>2 units/day)	0	9	0	16	0	14	1.00
Smokers	3	6	0	16	0	14	0.00
Physical activity (<30 min)	0	9	0	16	0	14	1.00
No calcium intake in diet	0	9	0	16	0	14	1.00
No sun	0	9	0	16	0	14	1.00
Humped parents	0	9	4	12	3	11	0.26
>40 years	9	0	16	0	14	0	1.00
Loss of more than 3 cm	0	9	4	12	3	11	0.26

Interventions: G1, only exercise; G2, exercise + calcium and vitamin D; G3, intense pace + calcium and vitamin D. Abbreviations: AI: after intervention; BI: before intervention.

**Table 2 ijerph-19-11215-t002:** T-score results before and after 2-years of intervention.

		*T*-Score BL	*T*-Score EI	*T*-Score BC		Time		Group		Time × Group	
Body Area	*n*	Mean (SD)	Mean (SD)	Mean (SD)	BC%	*F*	*p*-Value	*F*	*p*-Value	*F*	*p*-Value
Femoral neck						1.32	0.259	0.98	0.384	1.48	0.241
G1	9	−0.756 (0.746)	−0.811 (0.680)	−0.056 (0.229)	−7.3%						
G2	16	−0.786 (0.656)	−0.602 (0.708)	0.184 (0.430)	23.4%						
G3	14	−1.079 (0.830)	−1.014 (0.695)	0.064 (0.266)	6.0%						
Lumbar spine						4.51	0.041 ^2–3^	2.61	0.081	2.33	0.112
G1	9	−1.479 (0.254)	−1.577 (0.471)	−0.100 (0.630)	−6.8%						
G2	16	−1.913 (1.061)	−1.383 (0.920)	0.529 (0.785)	27.7%						
G3	14	−2.307 (0.793)	−2.000 (0.748)	0.307 (0.634)	13.3%						

Interventions: G1, only exercise; G2, exercise + calcium and vitamin D; G3, intense pace + calcium and vitamin D; Abbreviations: BL, baseline; EI, end of intervention; BC, baseline change; BC, %; and T-score (EI), T-score (BL)/T-score (BL). 2–3, differences between group 2 and group 3.

**Table 3 ijerph-19-11215-t003:** Number of participants experiencing fall and/or fracture events.

	G1 (*n* = 9)	G2 (*n* = 16)	G3 (*n* = 14)	*χ* ^2^	*p*-Value	Cramer V	*p*-Value
Falls (last 2 years)				3.21	0.201	0.29	0.213
Fallers	0 (0.0%)	1 (6.3%)	3 (21.4%)				
Non-fallers	9 (100%)	15 (93.8%)	11 (78.6%)				
Fractures (last 2 years)				6.77	0.034	0.42	0.037
Participants with fracture	1 (11.1%)	1 (6.3%)	6 (42.9%)				
Participants with no fracture	8 (88.9%)	15 (93.8%)	8 (57.1%)				

Interventions: G1, only exercise; G2, exercise + calcium and vitamin D; and G3, intense pace + calcium and vitamin D.

## Data Availability

The data that support the findings of this study are available from the corresponding author, [C.G.-G.], upon reasonable request.

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
