# Peer review of "Effects of Three Interventions Combining Impact or Walking at Intense Pace Training, with or without Calcium and Vitamin Supplements, to Manage Postmenopausal Women with Osteopenia and Osteoporosis"

_ijerph, 2022, doi:10.3390/ijerph191811215_

Round 1
Reviewer 1 Report (Previous Reviewer 2)
A more in-depth and detailed description of type of nutritional recommendation or program could be interesting.
Author Response
Dear reviewer 1,
Thank you for your feedback,
As for your comment: A more in-depth and detailed description of type of nutritional recommendation or program could be interesting.
We completely agree with the reviewer that this is a fundamental aspect in this type of patient. Item 18 of the Osteoporosis Foundation test that we pass on to patients emphasizes this aspect, which is why the consumption of dairy products, etc., is recommended. However, we did not give a detailed program of recommendations, although this was not the objective of this study. Undoubtedly, this aspect is of interest in future studies, so, given your comment, we have decided to indicate it at the end of the discussion, to refer to possible limitations and future lines of research.
Once again, thank you for your time and review work,

Reviewer 2 Report (New Reviewer)
- The study attempts to evaluate the effects of three interventions on BMD on postmenopausal women with affected bone density. The introduction is comprehensive and well written.
- The study design has some minimal flaws. the number of cases with osteoporosis, respectively osteopenia is not clear, it is not clear if they were equally distributed or the results may be affected by higher prevalence of osteoporosis in some groups or more severe cases in one group
- In the following sentence: line 100 "Women who were supplemented with medications for osteoporosis other than calcium or vitamin D", the word supplemented is not appropriate
- Here it is also strange that patients with known osteoporosis were not receiveing specific treatment such as bisphosphonates.. Please explain more. Current guidelines do recommend treatment
Author Response
Dear reviewer 2,
Thank you for your feedback,
We have tried to address your suggestions to increase the quality of the manuscript,
Specifically:
- The study attempts to evaluate the effects of three interventions on BMD on postmenopausal women with affected bone density. The introduction is comprehensive and well written.
- Thank you for your comment.
- The study design has some minimal flaws. the number of cases with osteoporosis, respectively osteopenia is not clear, it is not clear if they were equally distributed or the results may be affected by higher prevalence of osteoporosis in some groups or more severe cases in one group.
- We have decided to upload supplementary material that shows a summary together with the diagnosis of each included participant that had either 0=1 normal, 1=osteopenia or 2=osteoporosis at lumbar spine and femoral neck. All the raw data for analyses has been uploaded too. The statistical analyses were conducted accounting for basal differences and time changes from baseline scores, to prevent the issue raised by the reviewer. We hope this helps to clarify this issue.
- In the following sentence: line 100 "Women who were supplemented with medications for osteoporosis other than calcium or vitamin D", the word supplemented is not appropriate
- Indeed, we have reviewed the English and changed the word by ‘prescribed’. Thank you for noticing
- Here it is also strange that patients with known osteoporosis were not receiveing specific treatment such as bisphosphonates. Please explain more. Current guidelines do recommend treatment
- The reviewer is right. But please consider that the physician was the one who gave the treatment to be followed during the study. The patient's history was accessed and the sample was eligible according to potential comply with the inclusion criteria,
Once again, thank you for your time and review work,
DIAGNOSIS
|
|
Group 1 |
Group 2 |
Group 3 |
|
DX femoral |
6 normal 3 osteopenia 0 osteoporosis |
11 Normal 5 Osteopenia 0 Osteoporosis |
5 Normal 8 Osteopenia 0 Osteoporosis |
|
DX lumbar spine |
0 Normal 9 Osteopenia 0 Osteoporosis |
2 Normal 10 Osteopenia 4 Osteoporosis |
1 Normal 8 Osteopenia 5 Osteoporosis |
0 Normal, 1 Osteopenia, 2 Osteoporosis
|
GROUP |
DX femoral |
DX spine |
|
1 |
0 |
1 |
|
1 |
1 |
1 |
|
1 |
0 |
1 |
|
1 |
2 |
1 |
|
1 |
2 |
1 |
|
1 |
0 |
1 |
|
1 |
0 |
1 |
|
1 |
0 |
1 |
|
1 |
0 |
1 |
|
1 |
6 normal 3 osteopenia 0 osteoporosis |
0 Normal 9 Osteopenia 0 Osteoporosis
|
|
2 |
0 |
1 |
|
2 |
0 |
1 |
|
2 |
0 |
2 |
|
2 |
0 |
1 |
|
2 |
0 |
2 |
|
2 |
0 |
2 |
|
2 |
1 |
0 |
|
2 |
1 |
0 |
|
2 |
1 |
1 |
|
2 |
0 |
1 |
|
2 |
0 |
1 |
|
2 |
0 |
1 |
|
2 |
0 |
1 |
|
2 |
0 |
1 |
|
2 |
1 |
2 |
|
2 |
1 |
1 |
|
2 |
11 Normal 5 Osteopenia 0 Osteoporosis |
2 Normal 10 Osteopenia 4 Osteoporosis |
|
3 |
0 |
2 |
|
3 |
1 |
2 |
|
3 |
1 |
2 |
|
3 |
0 |
1 |
|
3 |
1 |
2 |
|
3 |
1 |
1 |
|
3 |
1 |
1 |
|
3 |
1 |
1 |
|
3 |
0 |
1 |
|
3 |
0 |
2 |
|
3 |
0 |
1 |
|
3 |
1 |
0 |
|
3 |
1 |
1 |
|
3 |
1 |
1 |
|
3 |
5 Normal 8 Osteopenia 0 Osteoporosis |
1 Normal 8 Osteopenia 5 Osteoporosis |

This manuscript is a resubmission of an earlier submission. The following is a list of the peer review reports and author responses from that submission.
Round 1
Reviewer 1 Report
Dear authors,
Please, see my comments below:
1. The abstract is very confusing and must be restructured. Attain to the findings, please.
2. "Regarding group allocation, it was not possible to randomize the entire sample in origin, because the calcium and vitamin D supplements were medically prescribed for the participants before they took part in the research, and this factor could not be controlled." This statement must be explained in detail. How the calcium and vit D supplements intake previously to your assessments could change the outcomes?
3. I could not notice the sample size calculation. It seems to me that the sample is too small and subject to type 2 errors over the findings' interpretation. Please, explain and make a statement about that.
4. Due to a very small sample, you might opt to use non-parametric assessments. I would also question about the asumption checks. It would be very useful if you provide the p-values and the Q-Q plots.
5. You see, the reported data sometimes does not match with the asumption checks. The SD is very close the the reported mean. For transparency, deposit the raw data in an online repository, such as Mendeley Data.
6. I strongly believe that the conclusions must be moderated. Despite the study design (RCT), no inferences could take place without a sample size calculation.
7. Aside from the above mentioned issues, the novelty of the present study is not clear to me. There are several studies mentioning the usefulness of exercising and supplementing to prevent future bad outcomes. Thus, please, clarify what the present study adds to the current available evidence.
